# Functional Availability of ON-Bipolar Cells in the Degenerated Retina: Timing and Longevity of an Optogenetic Gene Therapy

**DOI:** 10.3390/ijms222111515

**Published:** 2021-10-26

**Authors:** Jakub Kralik, Sonja Kleinlogel

**Affiliations:** Translational Optogenetics Group, Department of Physiology, University of Bern, Bühlplatz 5, 3012 Bern, Switzerland; jakub.kralik@unibe.ch

**Keywords:** optogenetic gene therapy, mGluR6, Opto-GPCRs, vision restoration, retinal degeneration, ON-bipolar cells, ganglion cells, multi-electrode array recordings, *rd1* mouse line, micro ERGs

## Abstract

Degenerative diseases of the retina are responsible for the death of photoreceptors and subsequent loss of vision in patients. Nevertheless, the inner retinal layers remain intact over an extended period of time, enabling the restoration of light sensitivity in blind retinas via the expression of optogenetic tools in the remaining retinal cells. The chimeric Opto-mGluR6 protein represents such a tool. With exclusive ON-bipolar cell expression, it combines the light-sensitive domains of melanopsin and the intracellular domains of the metabotropic glutamate receptor 6 (mGluR6), which naturally mediates light responses in these cells. Albeit vision restoration in blind mice by Opto-mGluR6 delivery was previously shown, much is left to be explored in regard to the effects of the timing of the treatment in the degenerated retina. We performed a functional evaluation of Opto-mGluR6-treated murine blind retinas using multi-electrode arrays (MEAs) and observed long-term functional preservation in the treated retinas, as well as successful therapeutical intervention in later stages of degeneration. Moreover, the treatment decreased the inherent retinal hyperactivity of the degenerated retinas to levels undistinguishable from healthy controls. Finally, we observed for the first time micro electroretinograms (mERGs) in optogenetically treated animals, corroborating the origin of Opto-mGluR6 signalling at the level of mGluR6 of ON-bipolar cells.

## 1. Introduction

It is estimated that approximately 43 million people worldwide suffer from blindness, with an expected increase of this number to 61 million by 2050 [1]. From these, approximately 2.4 million suffer from retinitis pigmentosa (RP) [2], and 3.8 million from age-related macular degeneration (AMD) [3], both being currently untreatable causes of progressive photoreceptor degeneration. Given the prolonged life expectancy [4] and prevalence of blindness in the aging population [5], the exploration of possible treatment strategies is essential not only to increase the quality of life in affected patients, but also to alleviate economic burdens linked to blindness [6,7,8]. Indeed, in the last decade the effort of many research groups propelled the field of vision restoration into new, previously unimaginable dimensions [9,10,11]. In the early stages of retinal degeneration, with a still relatively intact retinal architecture, gene replacement [12,13,14,15] and pharmacology [16,17,18] are employed. In the case of complete photoreceptor loss, the treatment strategies include electronic prostheses [19,20], stem cells and cell transplantation [21], photo-switchable ligands [22,23], and optogenetics [24,25,26,27]. Fundamental for all of these strategies is the intactness of the remaining retinal cells, the most critical being the layer of retinal ganglion cells (RGCs) that facilitates the transmission of signals from the retina to higher visual centers.

Optogenetics holds significant therapeutical potential. Recently, Sahel et al. [28] reported visual function recovery in blind patients after adeno-associated virus-mediated (AAV) ChrimsonR [29] optogenetic gene therapy. Not only is this a major milestone in the application of optogenetics, but it also represents one of the more design-wise basal optogenetic gene therapies with the targeting of RGCs and expressing microbial light-sensitive channels. Bipolar cells undoubtedly represent an attractive alternative target given their potential to restore inner retinal processing, which is impossible to achieve via RGCs targeted therapy, since it effectively transforms the entire cellular population into uniform light detectors.

Advances in the targeting of ON-type bipolar cells (OBCs) by both the development of specific promoters [30,31] as well as synthetic AAVs [32,33,34,35] have effectively enabled OBC-targeted optogenetic vision restoration. To move the “natural” vision restoration even further, one should aim for the activation of the most upstream signalling component. In the case of the OBCs, it is the metabotropic glutamate receptor 6 (mGluR6) that responds to glutamate released by photoreceptors. The modulation of mGluR6 activity subsequently modifies the signalling cascade, resulting in the activation/deactivation of the non-specific cation channel TRPM1 [36,37]. Indeed, we previously reported the use of designer chimeric opsin Opto-mGluR6 that couples the light sensitive domains of melanopsin and the intracellular domains of mGluR6. Coupled with the specific targeting of OBCs, we were able to effectively modulate the natural mGluR6 cascade and elicit robust light responses [38].

Opto-mGluR6-based gene therapy has the theoretical potential to outperform classical microbial-opsin-based optogenetic gene therapies [11,25,39]. On the other hand, Opto-mGluR6 gene therapy requires (1) the survival of OBCs in RP, (2) the preservation of the mGluR6 signalling pathway, and (3) the preservation of inner retinal signalling. The first noticeable effect of RP on bipolar cells is the aberrant connections they form with the remnants of photoreceptors, followed by dendritic retraction in the later stages of RP [40,41,42]. Focusing on OBCs, apart from gross anatomical changes, more subtle alterations take place. It has been reported that components of the mGluR6 signalling cascade become (1) insensitive [43], (2) relocalized [44], and (3) downregulated [45]. Moreover, hyperactivity of the retinal network is present in the mouse model of RP, originating from the coupling of cone OBCs and AII amacrine cells [46]. On the other hand, a recent transcriptomic analysis of bipolar cells in retinal degeneration [47] suggests the viability of bipolar cells as targets for optogenetic vision restoration. This notion is further supported by research findings that indeed present the functionality and preservation of mGluR6 signalling in RP both by direct glutamate activation [48] as well as introduction of mammalian Opto-GPCRs into these cells [31,38,49,50,51].

In this article, we want to evaluate the functional performance of Opto-mGluR6-treated blind *rd1* animals using multi-electrode arrays (MEAs) that enable us to record electrical signals from RGCs, the very last cells of the photo-signalling cascade of retina. The *RD1* mouse model mimics the course of RP, the most common inherited disease of the retina [52]. Specifically in the C3H *rd1* mice we employed, the course of the disease is characterized by the progressive degeneration of photoreceptors and relatively long preservation of the inner retinal layers [53,54]. One of the central questions regarding optogenetic gene therapy in RP is the feasibility of its usage in later the stages of retinal degeneration and remodeling. The C3H *rd1* model of RP exhibits several major hallmarks of progressive degeneration (Appendix A), with the first being the death of rods at about 4 weeks of age [55,56]. The rod-less retina retains an outer segments of cones up to 12 weeks of age, with the degenerative process peaking at week 7–8 [56]. Cone cell bodies, however, are more resistant to RP, and remain functional for prolonged periods of time, evident by the fact that the retinal ganglion cell (RGC) light evoked activity up to 24 weeks of age [54]. Worth mentioning is the 16 weeks point in time, when a drop in optokinetic reflex (OKR) as well as cone cell bodies count was observed [54]. Only from 25 weeks of age onwards can we talk about truly blind animals devoid of the influence of photoreceptors. For this very reason, all of our experiments were performed on animals aged P > 175 (>25 weeks). After this time point, further retinal remodeling was reported [41], however, the impact of such remodeling on the use of OBC-targeted optogenetic gene therapy in the later stages of RP was never fully explored.

Here, we report four important findings: (1) rAAV-mediated Opto-mGluR6 optogenetic gene treatment is not only long-lived, but also sufficient in vision restoration in advanced stages of retinal degeneration; (2) apart from robust spiking restoration, Opto-mGluR6 gene therapy is also capable of restoring local field potentials (LFPs) observed for the very first time in optogenetically treated blind retina; (3) Opto-mGluR6 gene therapy restores basal retinal firing rates to levels not significantly different from healthy C57BL/6 animals; (4) the mGluR6 signalling pathway is functional and responsive to both pharmacological and optogenetic activation.

## 2. Results

### 2.1. Opto-mGluR6 Optogenetic Gene Therapy in Murine Blind Retains Long-Lived Functionality and Successfully Restores Light Sensitivity Even in Advanced Retinal Degeneration

The recombinant adeno-associated virus (rAAV), carrying the gene encoding Opto-mGluR6 and a far-red fluorescent protein (TurboFP635), was intravitreally injected (Figure 1A). Subsequently, we performed a functional evaluation of retinal light-signalling using multi-electrode array (MEA) recordings considering two time points: (1) the age at treatment administration and (2) the age at treatment evaluation. We performed MEA recordings from retinal flat-mounts of treated *rd1* mice (Figure 1B) of >25 weeks of age, when all photoreceptor-driven light responses are gone (Appendix A) [54], to mimic the extensive photoreceptor degeneration seen in end-stage patients suffering from RP.

Given the fact that melanopsin-driven light responses originating from intrinsically photosensitive retinal ganglion cells (ipRGCs) seem to be upregulated in degenerated mouse retina [57], we focused solely on light responses that peaked early on (during the light flash), which is uncharacteristic for melanopsin [58]. Moreover, the presence of such responses hints toward the preservation of a functional inner retinal network, given the nature of the mGluR6 signalling [36,37]. We observed robust increases in the firing rates of RGCs upon light stimulation (Figure 1C; 470 nm; 5 × 10^14^ photons/cm^2^/s; 1 s) and considered the peak firing rates in our analysis.

We did not observe any significant changes in the peak firing rates (Figure 1D; Appendix A) in retinal explants functionally evaluated between 25–30 weeks of age (N = 29 retinal explants; n = 132 cells; 56.3 ± 2 8.3 Hz), 30–35 weeks (N = 25 retinal explants; n = 91 cells; 58.7 ± 28.7 Hz), 35–40 weeks (N = 12 retinal explants; n = 43 cells; 67.4 ± 28.6 Hz), and >60 weeks (N = 12 retinal explants; n = 63 cells; 64.4 ± 42.8 Hz). This observation indicated the minimal influence of progressive retinal degeneration on Opto-mGluR6-derived light responses in advanced age. The presence and proper localization of the Opto-mGluR6 construct in treated *rd1* animals, even in advanced age, was verified using immunohistochemistry on vertical cryosections (Appendix A). Although the age at which the animals were tested did not seem to play a significant role, there was still a possible influence of point in time on treatment administration. As retinal degeneration is a progressive process characterized by several major events (Introduction; Appendix A), we compared groups of mice treated at 2–4 weeks (N = 12 retinal explants; n = 58 cells; 62.9 ± 30.3 Hz), 4–8 weeks (N = 5 retinal explants; n = 22 cells; 59.6 ± 25.7 Hz), 8–12 weeks (N = 17 retinal explants; n = 67 cells; 53.3 ± 24.9 Hz), 12–16 weeks (N = 17 retinal explants; n = 48 cells; 70.8 ± 37.8 Hz), 16–25 weeks (N = 21 retinal explants; n = 101 cells; 56.6 ± 28.7 Hz), and >25 weeks (N = 10 retinal explants; n = 40 cells; 58.3 ± 42.9 Hz). Our results show (Figure 1E; Appendix A) no significant differences between tested groups, proposing the viability of optogenetic intervention even in the later stages of retinal degeneration. Lastly, we compared peak responses of RGCs originating from animals evaluated at a very late time point (aged >60 weeks), but treated at different time points (Figure 1F). We did not observe any significant differences between animals older than 60 weeks injected at a young age (<15 weeks; N = 5 retinal explants; n = 24cells; 74.6 ± 40.4 Hz) and old age (>50 weeks; N = 7 retinal explants; n = 39 cells; 58.2 ± 43.5 Hz).

Together, our results suggest that an OBC-targeted gene therapy is not only long-lasting, but also a viable treatment strategy in the very end-stage of RP. It is remarkable that, given the harsh nature of retinal degeneration characteristic for *rd1* animals, we did not observe any significant functional changes.

### 2.2. Micro ERGs Observed for the Very First Time in Optogenetically Treated Murine Retina

One of the most commonly used assessments of the preserved vision in patients with suspicion of retinal impairment is the use of electroretinogram recordings (ERG) that evaluate the gross function of the retina [59,60]. In the case of heterogeneous photoreceptor degeneration, spatially restricted areas are functionally tested by focal ERGs. MEAs were proven to be effective in recording so called micro ERGs (mERGs) from retinal explants that are effectively equivalent to in vivo focal ERGs [61]. A hallmark of RP, both in humans and mice, is the absence of ERGs in later stage degeneration [54].

Here, we report for the very first time the appearance of mERGs upon optogenetic stimulation (470 nm; 5 × 10^14^ photons/cm^2^/s; 1 s) in retinas (N = 17 retinal explants) from treated blind *rd1* mice (Figure 2A). The uncharacteristic negative deflection of “b-wave” can be explained by the inner workings of the Opto-mGluR6 construct that causes the hyperpolarization of OBCs [38]. We compared mERG amplitude (Figure 2B; Appendix A) and latency (Figure 2C; Appendix A) in retinas (N = 10) from Opto-mGluR6-treated *rd1* mice upon repeated illumination, and found neither significant changes in latency nor amplitude, the latter supporting the bistable nature of the Opto-mGluR6 construct [38]. In retinas (N = 2) from dark-adapted C57BL/6 mice, the mERG amplitude decreased with repeated illumination (Figure 2D; Appendix A), whereas the latencies increased (Figure 2E, Appendix A), which we interpret as effects of photopigment bleaching and light adaptation. Untreated *rd1* animals did not exhibit any mERG-like light responses (data not shown).

To probe for the contributions of photoreceptors and OBCs to the mERG, we applied standardly used pharmacology [38,61]. In the case of the C57BL/6 retinas (Figure 2F), the characteristic negative deflection (a-wave) originating from photoreceptors persisted in the presence of the mGluR6 agonist L-(+)-2-amino-4-phosphonobutyric acid (L-AP4; 20 µM); however, as expected, the positive b-wave was abolished. In the case of the retinas of Opto-mGluR6-treated *rd1* mice (Figure 2G), the mERG was insensitive to L-AP4 proving it did not originate from photoreceptors. In order to confirm that the mERG observed in treated retinas is not simply an artifact of incomplete degeneration, we performed the same pharmacological evaluation in the retinas of young *rd1* mice (<4 weeks of age, Figure 2H). These animals exhibited a-wave-like mERGs, as previously reported by Fujii et al. [61], which disappeared in the presence of L-AP4. This simple observation confirmed that the origin of mERGs in Opto-mGluR6-treated rd1 retinas originates from the optogene. Figure 2I summarizes the changes in a-wave and b-wave amplitudes upon L-AP4 application. The C57BL/6 retinas showed a significant decrease in the a-wave (n = 33 traces; *p* = 0.025), whereas the b-wave of Opto-mGluR6-treated retinas (n = 13 traces) remained unaltered (*p* = 0.99; Mann-Whitney U-test). In untreated young *rd1* animals, the a-wave completely disappeared in the presence of L-AP4 (n = 9 traces; *p* < 0.0001).

These results clearly demonstrate that aside from light responses in the form of the RGC spiking, the Opto-mGluR6 optogenetic gene therapy is capable of restoring mERG-like responses in blind *rd1* retinas.

### 2.3. Preservation of mGluR6 Functionality and Effects of Opto-mGluR6 on Retinal Hyperactivity

Optogenetically treated *rd1* retinas generated robust and repeatable (5×; 470 nm; 5 × 10^14^ photons/cm^2^/s; 1 s) light-evoked increases in RGC firing rates, without response rundown (Figure 3A; Appendix A), confirming previously observed repeated mERGs (Figure 2B,C). During repeated light stimulation, we noticed an increase in the basal firing rate over time (Figure 3B). The statistical analysis of the basal firing rates before the 1st (5.5 ± 0.2 Hz) and the 5th (10.3 ± 0.3 Hz) light stimulation in treated *rd1* retinas revealed a significant increase (*p* < 0.00001, Figure 3C), which we also found in the retinas of seeing C57BL/6 mice (1st = 5.89 ± 0.78 Hz; 5th = 10.3 ± 1.3 Hz; *p* = 0.0061). We did not observe such differences in untreated, negative *rd1* controls (1st = 9.5 ± 0.6 Hz; 5th = 11.0 ± 0.6 Hz; *p* = 0.067). Another observation was the fact that the basal firing rate prior to any light stimulation did not significantly differ between the retinas of healthy C57BL/6 and Opto-mGluR6-treated animals (*p* = 0.23), hinting toward a positive effect of the Opto-mGluR6 gene therapy on the overall aberrant activity of the *rd1* retina.

To validate the involvement of the OBCs and the mGluR6 pathway in the enhancement of the basal firing rate in Opto-mGluR6-treated retinas, we applied the mGluR6 agonist L-AP4 (20 µM). (Figure 3D,E). We did not observe significant changes in the basal firing rates of C57BL/6 retinas before and after L-AP4 application (*p* = 0.13), although there was an approximately 50% increase in the average baseline activity, hinting toward the involvement of OBCs (and the inner retinal network) in the regulation of the basal RGC firing rate. On the other hand, we observed a significant increase in the basal firing rates of optogenetically treated *rd1* animals upon L-AP4 application (*p* < 0.00001). The same was true for untreated, negative *rd1* controls (*p* < 0.00001).

To further pursue the involvement of OBCs in the increase in the basal firing rate, we used additional pharmacological agents (Figure 3F, Appendix A—the AMPA/kainite receptor antagonist 6-cyano-7-nitroquinoxaline-2,3-dione (CNQX; 20 µM) and the NMDA antagonist DL-(-)-2-Amino-5-phosphonopentanoic acid (DL-AP5; 40 µM), which, in combination, effectively blocked the input from upstream neurons to RGCs. Moreover, we also applied meclofenamic acid (MFA; 80 µM), a potent gap junction blocker. The application of the antagonists as well as the gap junction blocker reduced the basal firing rate to normal levels, hinting toward its origin in AII amacrine cells and/or cone OBCs.

These observations confirm (1) that the OBCs, and in particular the mGluR6 signalling pathway, retain functionality in the degenerated *rd1* mouse retina and (2) that light-evoked increases in the basal firing rate in degenerated *rd1* retina are linked to mGluR6 cascade activation.

## 3. Discussion

The retina is a complex, multi-layered tissue that performs remarkable signal processing, extracting approximately 30 parallel channels of information from the visual scene [62,63,64]. The initiator cells of the visual cascade, the photoreceptors, act as light sensors that change their activity in response to light. In a similar manner, the RGCs, the very downstream elements of the visual cascade, do primarily act as the functional output of the retina, transforming chemical signals into electrical ones that are transmitted to the brain. Although the need for new vision restoration strategies is very pressing, given the complexity of the retina, it is a herculean venture to achieve. In particular, preserving the intricacies of visual processing is difficult but important; it is known that in the degenerating retina, not only the morphology of remaining retinal cells is changing [40,41,65], but also their activity [46] and protein expression or localization [44,45].

The use of optogenetics to battle retinal degeneration represents a viable union. To maximally preserve the inner retinal processing, and therefore maximize the restored visual output, bipolar cells represent promising targets. Optogenetic gene therapies with Opto-GPCRs go beyond, as they specifically couple to the mGluR6 cascade in OBCs that mediates light responses in healthy vision [31,38,50,51]. On the other hand, the success of an Opto-GPCR therapy depends on the survival of OBCs and the preservation of the mGluR6 cascade. It has been postulated that there is dysfunction or change in the localization of components of the mGluR6 cascade in retinal degeneration [40,41,43,44,45]. However, a substantial body of literature argues against this, and claims long-term stability of the mGluR6 signalling cascade [31,38,47,48,49,50,51,66]. To test the functional availability of OBCs in the degenerating retina, we set out to evaluate the restored functional output by Opto-mGluR6 expression using multi-electrode arrays. Firstly, we confirmed the presence of the Opto-mGluR6 proteins in treated *rd1* retinas (Figure 1B). We specifically recorded only from mice aged P > 175 to ensure the complete degeneration of any remnants of photoreceptors [54]. In accordance with previous findings [38], we observed robust increases in spiking frequencies upon light stimulation, confirming the functional preservation of signalling components necessary for signal propagation from Opto-mGluR6 in OBCs to the RGCs (Figure 1C). To specifically answer the question of if an OBC targeted optogenetic gene therapy still restores function in the late stages of retinal degeneration, we tested the retinas of differently aged treated *rd1* mice, going all the way to P = 448. We did not observe significant changes in functional output recorded from Opto-mGluR6 treated animals (Figure 1D), hinting toward the preservation of the necessary signalosome and inner retinal network up to the late stages of retinal degeneration. Such preservation includes not only components of the mGluR6 cascade, but also preserved connections from OBCs to RGCs, with AII amacrine cells being critical in the signal transmission. As we focused purely on the responses characterized by an increase in the firing rate during light stimulation, the presence of a sign-inversing component is most likely required; however, this needs further confirmation. Moreover, mGluR6 cascade activation may promote the preservation of OBC integrity. It is known from the literature that retinal remodeling is caused by deafferentation due to lack of input [42]; however, reintroducing such an input in OBCs may lead to the modulation and expression of mGluR6 components [15,67].

We next focused on the time of therapeutic intervention in *rd1* mice. The first group of animals was treated before 4 weeks of age, known to correspond to the peak of rod degeneration [55]. The second group was selected prior to the peak of cone outer segment degeneration that occurs in the 8th week of age and is complete by 12th week of age (3rd group) [54,56]. Two more groups (4th & 5th) were selected below the age of 25 weeks, by which the last remnants of cone cell bodies should disappear [41,54,56], and the sixth group beyond 25 weeks of age, when C3H *rd1* mice do not exhibit any photoreceptor-driven light responses [54]. Remarkably, we did not observe significant changes in restored retinal light sensitivity (Figure 1E), pointing toward the theoretical feasibility of successful OBC-targeted optogenetic treatment, even in patients with advanced retinal degeneration. In the last set of age-related experiments, we focused on groups older than P > 420 that differed only in the time points of therapeutic intervention (Figure 1F), which confirmed the consistency and longevity of vision restoration independent of the point of intervention. Previous reports [66] showed the longevity of optogenetic treatment targeted to RGCs for extended periods of time, but this seems to be the first OBC-targeted optogenetic gene therapy where such a parameter was evaluated. Moreover, the functionality in Opto-mGluR6-based gene therapy in advanced age supports the notion of the long-term preservation of inner retinal layers and their signalling elements.

Electroretinogram (ERG) recording is a standard clinical procedure used to evaluate retinal function [59,60], and ERGs elicited by Opto-mGluR6 were previously shown in the *rd1* retina [38]. We showed here, for the very first time, the focal counterpart, mERGs, in optogenetically treated retinas (Figure 2A). In line with previous reports [38], the pharmacological blockade of connections from photoreceptors to ON-bipolar cells with L-AP4 did not abolish the b-wave in the Opto-mGluR6-treated *rd1* retina, (Figure 2G) which is in stark contrast to the typical a-wave-like responses obtained from young *rd1* retinas still possessing photoreceptors that disappear in the presence of L-AP4 [61]. This confirmed that the origin of the electronegative b-wave in treated *rd1* mice is triggered by Opto-mGluR6, and originates in the OBCs.

We observed that the basal firing rate in treated animals increased with repeated light stimulation. Increased spontaneous firing rates are characteristic for the *rd1* mouse model, with frequencies typically in the range of 5–15 Hz, and believed to originate from the coupling of AII-amacrine cells and cone OBCs, being linked to pathologically hyperpolarized cone OBCs [46,68]. Light activation of Opto-mGluR6 expressed in OBCs also leads to their hyperpolarization by closure of non-specific cation TRPM1 channels [37], and may thus enhance spontaneous firing rates [68]. Pharmacology indeed confirmed the involvement of mGluR6 signalling cascade in the rd1 mouse model of RP in the increase in the basal firing rate. The use of the gap junction blocker MFA as well as DL-AP5 and CNQX hinted toward the origin of this hyperactivity being cone OBCs and/or AII amacrine cells, which is in agreement with previous observations [46].

Lastly, we also observed reduced spontaneous activity in dark adapted Opto-mGluR6 treated *rd1* animals, which did not differ significantly from healthy *C57BL/6* animals. We cannot claim that this is linked specifically to Opto-mGluR6 construct per se, but rather an Opto-GPCR input in the OBCs. As mentioned before, the introduction of input into bipolar cells seems to be imperative in the preservation of their functionality, as the localization and expression of OBC signalling components seems to be plastic and prone to modulation [15,67].

In summary, (1) OBC-targeted Opto-mGluR6 gene therapy remains functional for extended periods of time and is also sufficient in functional vision restoration in later stages of retinal degeneration; (2) for the first time, Opto-mGluR6 gene therapy demonstrated mERGs in optogenetically treated *rd1* retina; (3) the presence of Opto-mGluR6 gene therapy decreases retinal hyperactivity typical for the *rd1* mouse model of RP; (4) the mGluR6 signalling pathway retains functionality and responsiveness to both pharmacological and optogenetic activation.

## 4. Materials and Methods

### 4.1. DNA & Viral Constructs

Opto-mGluR6 carrying rAAV vectors were produced as previously described [38]. Plasmid pAAV-770En_454P(hGRM6)_Opto-mGluR6_IRES_TurboFP635_WPRE_BGH was created as previously described elsewhere [31], but with Opto-mGluR6 optogene instead of the *Opn1mw*. Viral vectors were produced in HEK293 cells by the triple plasmid co-transfection method, and empty virions were removed by density purification over an iodixanol gradient. The 40% iodixanol fraction was subsequently buffer exchanged by column chromatography over a 5 mL HiTrap heparin affinity column (Sigma-Aldrich). We packaged viral vectors with capsids AAV2(7m8) [32] as described in detail elsewhere [35]. The titers were all between 1 × 10^12^ to 5 × 10^13^ genome copies per mL and stored at −80 °C.

### 4.2. Animals: AAV Transduction & Maintenance

The experiments were performed on the C3H/HeOuJ retinal degeneration (*rd1*) (N = 43 animals), and the C57BL/6J wild-type (N = 7 animals) mouse lines were purchased from Jackson Laboratory (Bar Harbor, ME, USA). The animals were maintained under a standard 12-h light-dark cycle.

For the purpose of intravitreal (IV) injections, *rd1* mice (N = 36 animals) were anesthetized by an intraperitoneal injection of 100 mg/kg ketamine and 10 mg/kg xylazine. The pupil of the right eye was dilated with a drop of 10 mg/mL atropine sulphate (Théa Pharma). We then punctured the dorsal sclera approximately 1 mm from the corneal limbus using an insulin needle. The insulin needle was removed and a 33G blunt needle was maneuvered through the pre-made hole to the back of the eye (RPE injection kit from World Precision Instruments). We then injected 2.5 μL of the rAAV vector solution and waited for 2 min before retracting the injection needle form the eye. The second eye was subsequently injected using the same procedure. Following surgery, an antibiotic eye lotion (Isathal from Dechra Veterinary Products) was applied to the eyes to prevent infection and drying of the cornea. The injected mice were kept with enhanced lighting provided with a Philips HF3319 daylight lamp (10,000 lux) positioned 50 cm from the cage to guarantee sufficient light levels for the activation of ectopic opsins.

### 4.3. Immunohistochemistry

We fixed the mouse retinas after the enucleation and removal of the cornea and lens for 30 min in 4% paraformaldehyde (PFA), and subsequently cryo-protected them in graded sucrose solutions. For cryosectioning, retinas were embedded with Tissue-Tek O.C.T. Compound (Sakura Finetek, Horgen, Switzerland ) in cryomolds and subsequently frozen using liquid-nitrogen-cooled 2-methylbutane. Frozen blocks were cut into vertical cryo-sections (thickness: 15 μm) on cryostat (Leica Biosystems, Muttenz, Switzerland), and mounted on SuperFrost Ultra Plus Adhesion slides (Thermo Scientific, Basel, Switzerland). We used 2xNGS (Sigma-Aldrich, G9023, Buchs, Switzerland) (6% normal serum, 2% BSA, and 0.3% Triton X-100 in 1× PBS) and prepared antibody solutions in 1× blocking solution (mixed 1:1 with 1× PBS). The following primary antibodies were used: rabbit anti-TurboFP635 (1:500; Evrogen, AB234, Zurich, Switzerland) and mouse anti-Gαo(1:750; EMD Millipore, MAB3073, Schaffhausen, Switzerland). The secondary antibodies used were goat anti-rabbit Alexa 488 (1:400; Invitrogen, A11008, Basel, Switzerland) and goat anti-mouse CY3 (1:400; Invitrogen, A10521, Basel, Switzerland). After 1 h of blocking, we incubated slides in primary AB solution overnight at 4 °C, and subsequently in secondary AB solution for 2 h at room temperature. Finally, we mounted the stained samples on glass slides with a Fluoromount mounting medium (Sigma-Aldrich, F4680). Micrographs were taken on a Zeiss Laser Scanning Microscope 510 (Zeiss, Feldbach, Switzerland). The processing of image stacks was done using ImageJ (version 1.53m, Rasband WS, United States National Institutes of Health, Bethesda, Maryland, USA).

### 4.4. Multi-Electrode Array Recordings

Mice were dark adapted for 60 min and subsequently sacrificed using isoflurane and cervical dislocation. Following enucleation, eyes were dissected under dim red light conditions in an Ames’ medium (Sigma-Aldrich) that had been oxygenated for at least 60 min prior the procedure with carbogen (95% O_2_/5% CO_2_). The retinas were placed on multi-electrode arrays (60MEA200/30iR-Ti; Multi Channel Systems MCS GmbH) coated with Corning™ Cell-Tak Cell and Tissue Adhesive (Corning, Wiesbaden, Germany), with the ganglion cells facing towards the electrodes. The MEA was placed into the MEA recording device (MEA2100-System; Multi Channel Systems MCS GmbH, Reutlingen, Germany) positioned on a stage of a Zeiss Axioskop coupled to a pE2 light stimulator (precisExcite, CoolLED, Andover, United Kingdom) connected to an oscilloscope (Tektronix TDS210, Tektronix, Berkshire, UK) and signal generator (ELV TIG7000, ELV, Leer, Germany). Perfusion with oxygenated Ames’ medium (Sigma-Aldrich) was maintained at 5 mL × min^−1^. The temperature was maintained at 34 °C. After the placement of the MEA into the recording device, the retina was perfused with oxygenated Ames’ medium for 30 min in darkness. Light stimulation (unless stated otherwise: 465 nm, 5 × 10^14^ photons.cm^−2^ s^−1^) was delivered through a 5× objective positioned above the MEA recording device. Recorded signals were collected, amplified, and digitized at 25 kHz using MCRack software (version 4.6.2, Multi Channel Systems MCS GmbH, Reutlingen, Germany).

### 4.5. Spike Analysis

Filtering was performed on recorded signals using a 2nd order Butterworth high-pass filter with a cut-off frequency of 200 Hz. Action potentials were defined as electrical activity below 3.5–5 SDs of baseline activity and set specifically for each recording electrode based on the baseline noise. Subsequently, spike cutouts recorded by each electrode were sorted into single cell traces using Offline Sorter (version 4.6.0, Plexon, Dallas, TX, USA). Time points of single cell spike occurrences were extracted from the software for offline analysis using Matlab (version R2020b, MathWorks, Natick, MA, USA). For annotating cells as light responsive, we used two parameters, with the requirement of fulfilling both: (i) Threshold (TR) defined as a change in firing rate (baseline + 5*SD), between the average frequency prior to light stimulation and at least 1 time bin during or after the light flash (adopted and modified from De Silva et al. [66]), and (ii) using a light response index (LRI) [51]. We defined LRI  =  (maximal firing rate during/after stimulus − average firing rate before light stimulus)/(maximal firing rate during/after stimulus  +  average firing rate before light stimulus). Only cells with the respective LRI > 0.2 and TR crossing were considered light responsive. Moreover, when using light responsive cells, we focused primarily on the cells with a maximal response occurring during light stimulation.

### 4.6. LFP Analysis

Raw traces were filtered using a 2nd order Butterworth low-pass filter with a cut-off frequency of 20 Hz. Subsequently, traces were extracted and analyzed offline using Matlab (version R2020b, MathWorks, Natick, MA, USA). Latency was defined as time point of maximal negative deflection of LFP following light stimulation.

### 4.7. Statistical Analysis

Statistical analysis was performed using Matlab (MathWorks). Normal distributions were tested using a Kolmogorov-Smirnov test. As our data were not normally distributed, significance levels were determined using a Mann-Whitney ‘U’ test. In the figures, mean ± standard deviation (SD) is stated, unless indicated otherwise. The stars for statistical analysis indicate: * if *p* < 0.05, ** if *p* < 0.01, *** if *p* < 0.001 and **** if *p* < 0.0001. In case Bonferroni correction was used, significance levels and corresponding star annotations are stated in Appendix A.

## Figures and Tables

**Figure 1 ijms-22-11515-f001:**
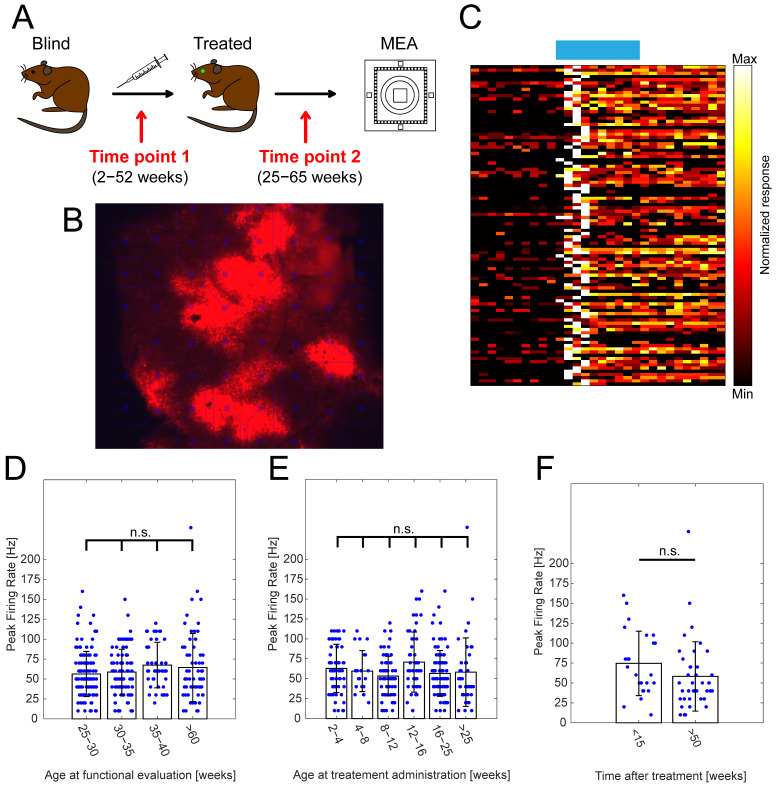
The effect of the point in time of treatment and the age at functional evaluation on RGC peak firing frequencies. (**A**) Schematic depicting experimental design of optogenetic treatment and functional evaluation using MEAs, highlighting two important time points: (1) age at treatment administration and (2) age at functional evaluation. (**B**) Visualization of TurboFP635 expression in Opto-mGluR6-treated rd1 retinal explant showing robust, albeit heterogenous transgene expression. (**C**) Exemplar heatmap showing normalized responses of RGCs (n = 100) in Opto-mGluR6-treated rd1 retinas to light stimulation (470 nm; 5 × 10^14^ photons/cm^2^/s; 1 s). (**D**) Peak firing rates of RGCs in Opto-mGluR6-treated rd1 retinas recorded at different ages of experimental animals (25–30 weeks, n = 132 cells; 30–35 weeks, n = 91 cells; 35–40 weeks, n = 43 cells; >60 weeks, n = 63 cells). We did not observe any significant differences between peak firing rates (Appendix A, α = 0.0083 after Bonferroni correction). Only cells that peaked during light stimulation were considered for analysis. (**E**) Peak firing rates of Opto-mGluR6-treated rd1 RGCs based on the time of treatment administration (2–4 weeks, n = 58 cells; 4–8 weeks, n = 22 cells; 8–12 weeks, n = 67 cells; 12–16 weeks, n = 48 cells; 16–25 weeks, n = 101 cells; >25 weeks, n = 40 cells). No significant differences were observed between experimental groups (Appendix A; α = 0.0167 after Bonferroni correction). (**F**) Comparison of peak firing rates of Opto-mGluR6-treated rd1 animals older than 60 weeks at the time of experiment. Difference between the two groups is the time of treatment administration (<15 weeks, n = 24 cells; >50 weeks, n = 39 cells). We did not observe any significant differences between the two experimental groups (*p* = 0.0512). Blue points represent individual cells.

**Figure 2 ijms-22-11515-f002:**
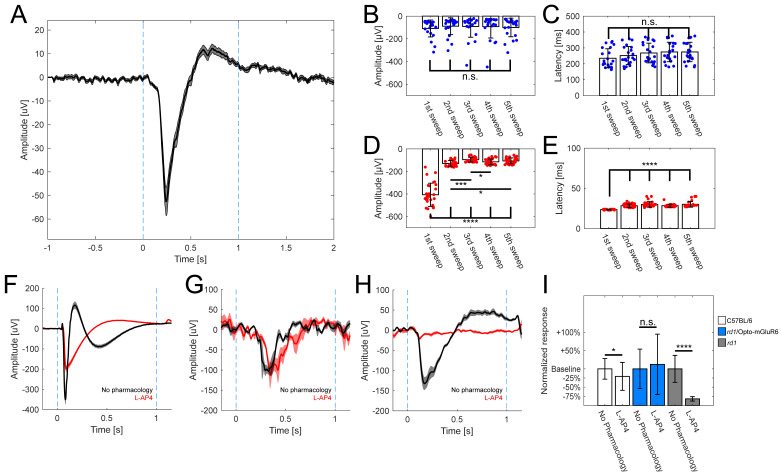
Opto-mGluR6 derived mERGs observed in *rd1*-treated retinas. (**A**) Averaged mERG derived from Opto-mGluR6-treated *rd1* retinas (n = 71 traces, mean ± s.e.m.). Light stimulation between blue dashed lines (470 nm; 5 × 10^14^ photons/cm^2^/s; 1 s), b-wave amplitude 93.3 ± 7.1 µV, latency 260 ± 7.1 ms. Comparison of mERGs amplitudes (**B**) and latencies (**C**) of Opto-mGluR6-treated *rd1* retinas (n = 22 traces) upon repeated light stimulation. We did not observe significant differences in amplitudes (Appendix A; α = 0.005 after Bonferroni correction) or latencies (Appendix A; α = 0.005 after Bonferroni correction) across light stimulations. Individual data points depicted as blue points. Comparison of amplitudes (**D**) and latencies (**E**) of mERGs of healthy C57BL/6 retinas (n = 29 traces) upon repeated light stimulation. We observed significant differences between the first light stimulation and all other subsequent stimulations in the amplitudes (Appendix A; α = 0.005 after Bonferroni correction; * *p* < 0.005, *** *p* < 0.0001, **** *p* < 0.00001) and latencies (Appendix A; α = 0.005 after Bonferroni correction; **** *p* < 0.00001) reflecting the transition from scotopic to photopic vision. Individual data points depicted as red points. Exemplar averaged mERGs derived from healthy C57BL/6 retinas (**F**, n = 18 traces), Opto-mGluR6 treated *rd1* retinas (**G**, n = 5 traces), *rd1* (<4 weeks of age) retinas (**H**, n = 9 traces) without any pharmacology (black trace), and L-AP4 (red trace). Data shown as mean ± s.e.m. In **F**, dark- (black) and light-adapted (red) a-waves remain in presence of L-AP4, which causes the b-wave to disappear (red trace). In **G**, the b-wave persists after L-AP4 application (red), indicative of its origin in the ON-BCs. In (**H**), L-AP4 abolishes the negative-wave in *rd1* controls. (**I**) Comparison of a-wave (C57BL/6, *rd1*) and b-wave (*rd1*/Opto-mGluR6) amplitudes. We observed significant difference in C57BL6 animals before and after application of L-AP4 (n = 33 traces; *p* = 0.028), in Opto-mGluR6 treated *rd1* animals we did not observe any effect of pharmacology (n = 13 traces; *p* = 0.99) and in the case of untreated young *rd1* control animals a-wave like responses completely disappeared in the presence of L-AP4 (n = 9 traces; *p* < 0.0001).

**Figure 3 ijms-22-11515-f003:**
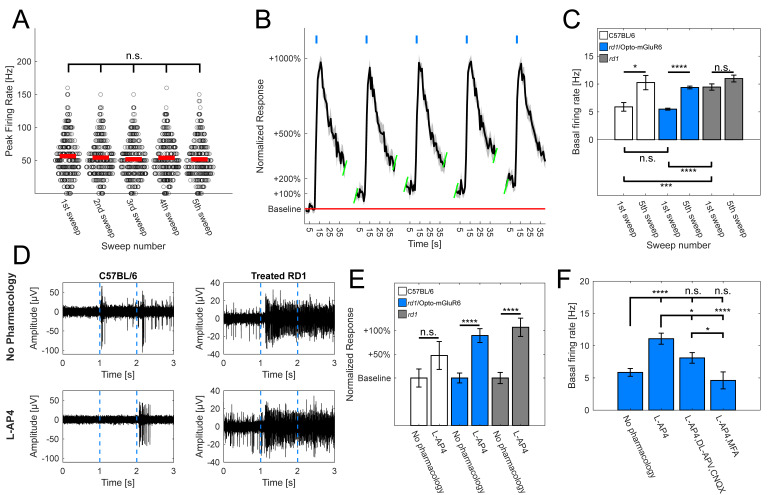
Effects of optogenetic and pharmacological stimulation of mGluR6 signalling pathway in *rd1* retina. (**A**) Comparison of peak firing rates derived from light responsive RGCs in Opto-mGluR6-treated animals (n = 291 cells). Peak firing rates did not differ significantly across multiple light stimulations (Appendix A; α = 0.005 after Bonferroni correction). Red lines depict means and dark circles individual cells. (**B**) Exemplar normalized light response spike-time histogram derived from RGCs of Opto-mGluR6 treated *rd1* retinas (n = 18 cells). Light flashes shown as blue rectangles. Peak firing rates do not change over time, however, there is a noticeable increase in the basal firing rate (red line). (**C**) Comparison of basal firing rates of healthy C57BL/6 animals (N = 7 retinal explants; n = 126 cells), treated *rd1* animals (N = 82 retinal explants; n = 2017 cells), and untreated negative *rd1* controls (N = 9 retinal explants; n = 313 cells) upon repeated light stimulation. There was a significant increase between basal firing rate prior to 1st and 5th light stimulation in C57BL/6 animals (1st = 5.9 ± 0.8 Hz; 5th = 10.3 ± 1.3 Hz; p = 0.0061; * *p* < 0.05, *** *p* < 0.001, **** *p* < 0.0001) as well treated *rd1* animals (1st = 5.49 ± 0.18 Hz; 5th = 10.26 ± 0.25 Hz; *p* < 0.00001). No such difference was observed in untreated *rd1* controls (1st = 9.46 ± 0.56 Hz; 5th = 10.99 ± 0.61 Hz; *p* = 0.067; α = 0.05, mean ± s.e.m.). Moreover, comparing basal firing rates before any light stimulation showed no significant differences between C57BL/6 and treated *rd1* animals (*p* = 0.23), but a significant increase in untreated *rd1* controls in comparison to both C57BL/6 (*p* < 0.00005) and Opto-mGluR6-treated *rd1* retinas (*p* < 0.00001; α = 0.0167 after Bonferroni correction; * *p* < 0.0167, *** *p* < 0.00033, **** *p* < 0.000033). (**D**) Exemplar raw traces of light responsive electrodes of C57BL/6 retinas (left) and treated *rd1* retinas (right), without any pharmacology (top) and with addition of L-AP4 (bottom). (**E**) Comparison of normalized baseline firing rates of healthy C57BL/6 animals, treated *rd1* animals, and untreated negative controls in absence or presence of L-AP4. The basal firing rate did not increase significantly in C57BL/6 retinas in the presence of L-AP4 (no pharmacology = 63 cells; L-AP4 = 33 cells; *p* = 0.13; * *p* < 0.05, *** *p* < 0.001, **** *p* < 0.0001), but increased significantly in Opto-mGluR6-treated *rd1* retinas (no pharmacology = 178 cells; L-AP4 = 173 cells; *p* < 0.00001) and untreated *rd1* control retinas (no pharmacology = 82 cells; L-AP4 = 74 cells; *p* < 0.00001). (**F**) Comparison of basal firing rates in Opto-mGluR6-treated *rd1* retinas prior to any pharmacological application (n = 178 cells; 5.9 ± 0.6 Hz), after application of L-AP4 (n = 173 cells; 11.1 ± 0.9 Hz), and the addition of either cocktail of CNQX and DL-APV (n = 144 cells; 8.1 ± 0.8 Hz) or MFA (n = 60 cells; 4.6 ± 1.3 Hz). We observed a significant decrease in the basal firing rate after additional pharmacology was added to L-AP4 in comparison to L-AP4 alone (Appendix A; α = 0.0083; * *p* < 0.0083, *** *p* < 0.000167, **** *p* < 0.0000167).

## Data Availability

The datasets used and/or analyzed during the current study are available from the corresponding author upon reasonable request.

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
