# Peer review of "Functional Availability of ON-Bipolar Cells in the Degenerated Retina: Timing and Longevity of an Optogenetic Gene Therapy"

_ijms, 2021, doi:10.3390/ijms222111515_

Round 1
Reviewer 1 Report
The authors here presented a very interesting and well-written paper. However, a few issues need to be addressed before publication:
- What many animals and retinas were used? The authors only mentioned the number of cells.
- Which strain of C57BL/6 was used? Where they tested for the rd8 mutation?
- Fig 1D, E and F please improve the lettering, it is difficult to read as it is. Also, add the age of the animals in timepoints 1 and 2 in the experimental scheme.
- Fig 3B, why the difference in the total number of cells between the different conditions?
- Please use retinas OR retinae throughout the text.
Reviewer 2 Report
Jakub Kralik and Sonja Kleinlogel submitted a paper describing the functional availability of ON-bipolar cells in the degenerated retina under optogenetic gene therapy. Through their experiments, several significant findings could be identified, including (1) rAAV mediated Opto-mGluR6 optogenetic treatment is long-lived and sufficient in vision restoration even in advanced stages of retinal degeneration; (2) optogenetic therapy is capable of restoring local field potentials (LFPs) in blind retina; (3) Opto-mGluR6 gene therapy restores basal retinal firing rates to levels not significantly different from healthy C57BL/6 animals and (4) mGluR6 signalling pathway is functional and responsive to both pharmacological and optogenetic activation. Overall this paper is well written, however, some minor concerns have to be answered.
1. In Fig 1 B , TurboFP635 expression in Opto-mGluR6 treated rd1 retinal explant showing robust, heterogenous gene expression. Why the expression pattern of Opto-mGluR6 in retina is heterogenous instead of homogenous after intravitreal injection?
2. Histopathological sections of the treated retinas should be provided to demonstrated the actual expression of Opto-mGluR6 gene located in the bipolar cell layers in the retina, and even in the advanced stage of retinal degeneration.
